# Identification of the CNGC Gene Family in Rice and Mining of Alleles for Application in Rice Improvement

**DOI:** 10.3390/plants12244089

**Published:** 2023-12-06

**Authors:** Xinchen Wang, Fengcai Wu, Jinguo Zhang, Yaling Bao, Nansheng Wang, Guohui Dou, Dezhuang Meng, Xingmeng Wang, Jianfeng Li, Yingyao Shi

**Affiliations:** College of Agronomy, Anhui Agricultural University, Hefei 230036, China; 22721752@stu.ahau.edu.cn (X.W.); 22720535@stu.ahau.edu.cn (F.W.); zhangjinguo@stu.ahau.edu.cn (J.Z.); valleybao2019@126.com (Y.B.); 1341489953@stu.ahau.edu.cn (N.W.); douguohui@stu.ahau.edu.cn (G.D.); 22721677@stu.ahau.edu.cn (D.M.) wangxingmeng@stu.ahau.edu.cn (X.W.); 22720528@stu.ahau.edu.cn (J.L.)

**Keywords:** rice, cyclic nucleotide-gated ion channel gene, geng-cds-haplotype (gcHap) diversity, modification

## Abstract

Cyclic nucleotide-gated ion channel (CNGC) gene regulation plays important roles in plant immune and abiotic stress response. Here, we identified 16 CNGC genes in rice (Oryza sativa). Then, we analyzed their chromosomal location, physicochemical properties, subcellular localization, gene functional interaction network, cis-acting elements, phylogenetic relationships, collinearity, expression in tissues under normal conditions and abiotic stresses, and geng-cds-haplotype (gcHap) diversity in 3010 gcHaps. As a result, *OsCNGC3* (*Os06g0527300*) was identified as a gene different from previous report, and *OsCNGC* genes were found to play important roles in rice population differentiation and rice improvement. Our results revealed their very strong differentiation between subspecies and populations, important roles in response to abiotic stresses, as well as strong genetic bottleneck effects and artificial selection of gcHap diversity in the modern breeding process of Xian (indica) and Geng (japonica) populations. The results also suggested that natural variations in most rice CNGC loci are potentially valuable for improving rice productivity and tolerance to abiotic stresses. The favorable alleles at the CNGC loci should be explored to facilitate their application in future rice improvement.

## 1. Introduction

Cyclic nucleotide-gated channels (CNGC) are widely present in plants and animals, playing important roles in their growth and development and response to stress. Cyclic nucleotides (CNMPs) are an important class of signaling molecules, such as 3′,5′-cyclic adenosine monophosphate (cAMP), which is an important component in signaling pathways in plants and animal life activities [1]. The CNGC family was first discovered in plants during the screening of a barley-pasteurized layer expression library for calmodulin (CaM) [2]. Sixteen members of the CNGC family have been identified in rice [3,4], among which four members have been cloned and verified. SSS1-D encodes *OsCNGC13*, a member of the CNGC family, which enhances fruit setting by promoting the growth of pollen tubes in the stylar tissue [5]. In Arabidopsis thaliana, *AtCNGC7* and *AtCNGC8* proteins are closely related to 74% amino acid sequence identity and play certain roles in pollen germination and male fertility [6]. *AtCNGC5*, *AtCNGC6*, and *AtCNGC9* are required for the structural growth of root hairs in *A. thaliana* [7]. *OsCNGC9* is a positive regulator of immune response (PTI), and *OsCNGC9* overexpression in rice transgenic lines showed significant enhancement of path-ogen-associated molecular patterns (PAMPs)-triggered immune response (PTI) and blast resistance include stronger calcium ion flux, higher burst of reactive oxygen species, and PTI-related gene expression levels [8], whereas in *A. thaliana*, *AtCNGC2* and *AtCNGC4* are positive regulators of PTI only at specific calcium concentrations (i.e., 1.5 mmol-L-1) [9]. In apple, *MdCNGC1* is probably a negative regulator of plant resistance to bacterial and fungal pathogens [10]; and in tomato, *SlCNGC1* and *SlCNGC14* inhibit rice stripe disease [11]. Moreover, overexpression of *OsCNGC9* was also found to significantly enhance rice resistance to low-temperature stress [12]. *OsCNGC14* and *OsCNGC16* positively regulate both heat and cold tolerance in rice, and their loss of function reduces or eliminates cytoplasmic calcium signaling induced by high or low-temperature stresses [13]. In Chinese jujube, a *ZjCNGC2*-mediated *ZjMAPK* cascade is involved in cold stress response [14]. In addition, *OsCNGC4*, *OsCNGC5*, and *OsCNGC8* may be associated with pollen development [4]. However, it remains to be explored how to target these CNGC genes for the improvement of crop yield and sustainability. Grain shape has an important impact on rice yield and its appearance, processing, cooking, and eating quality, thereby directly affecting the commercial value of rice. In addition, thousand-grain weight (TGW), a genetically stable trait, is another important factor affecting rice yield [8]. The factors that determine the TWG include grain length, grain width, and grain thickness [9]. Globally, different countries and regions have different preferences for rice quality traits. For example, people in South and Southeast Asia, southern China, the USA, and Latin America prefer long, fine grains with a fluffy and firm texture and medium amylose content. However, people in northern China, Korea, Japan, and parts of the Mediterranean area prefer short, round, soft, and sticky rice grains with a low amylose content [10,11].

Rice is one of the most important cereal crops in the world as well as a cereal model plant with high intraspecific genetic diversity [15,16]. In rice, more than 4600 genes have been cloned and functionally characterized at the molecular level (https://funricegenes.github.io/ (accessed on 26 September 2023)) [17]. The primary gene pool was found to have extremely high genomic diversity by sequencing 3010 different rice materials (3KRG) from 89 countries worldwide [18,19]. However, the rapid progress in functional and population genomic studies in rice has not yet been widely applied to the development of more efficient breeding techniques. This is due to the fact that there is incomplete information on the phenotypic effect(s) of cloned genes because most experiments were conducted in the laboratory instead of breeding target environments, and therefore, the effects of environmentally interacting genotypes and genetic backgrounds of most cloned genes on some important agronomic traits are largely uncertain [17]. Moreover, before spending much effort and funds to study a specific gene of interest, researchers should check whether similar alleles have been identified or fixed in commercial crop varieties; and if a gene has been studied for decades, it is unlikely to contribute to a sudden significant increase in yield [20]. Finally, due to the abundance of natural allelic variations at most loci in rice populations [19], it remains a great challenge to identify and mine desirable alleles from rice germplasm resources to improve specific target traits. Among *OsCNGC* genes, the functions of many genes still remain unknown. Therefore, it is time-consuming to understand the function of these genes by gene cloning, and a great challenge is to obtain the information about these genes needed for breeding without gene cloning.

In this study, we integrated a series of approaches including gene family identification and gene expression analysis with population genetic analysis of geng-cds-haplotype (gcHap) diversity in rice populations, developing an efficient strategy to provide important information on a large number of gene loci needed for future development of new breeding techniques of rice. 

## 2. Results

### 2.1. Genome-Wide Identification and Characterization of OsCNGC Genes

*OsCNGC* genes in the whole rice genome were identified using BLAST, HMM search, and literature analysis [3,4]. Finally, a total of 16 *OsCNGC* genes were identified in the genome. To characterize these *OsCNGC* genes, we further identified their physical and chemical features, including their protein length, molecular weight, instability coefficient, lipolysis index, hydrophilicity index, and subcellular localization. The 16 genes showed protein sequence lengths ranging from 243 to 773 amino acids, molecular weights ranging from 27,613.65 to 88,353.96 Da, and PI (theoretical) ranging from 8.02 to 10.04 with an average of 9.36 (Appendix A). Subcellular localization revealed that all 16 *OsCNGC* genes were localized to the plasma membrane or chloroplasts (Appendix A). The *OsCNGC* genes showed different physicochemical properties, indicating their different biological functions. Moreover, the 16 genes were distributed on all chromosomes except for chromosomes 7, 8, 10, and 11 (Figure 1a). To further reveal the evolutionary and structural characteristics of *OsCNGC* genes, we constructed a phylogenetic tree with 16 CNGC genes in rice (japonica), 20 CNGC genes in *A. thaliana*, 12 CNGC genes in maize, and 19 CNGC genes in Maoyamoya fruit, which could be classified into groups Ⅰ, Ⅱ, Ⅲ, and Ⅳ according to the distance of the evolutionary relationship, among which group Ⅳ could be subdivided into Ⅳ-A and Ⅳ-B subgroups (Figure 1b).

*OsCNGCs* had 14 to 19 motifs, and all contained CAP_ED structural domains, while the structural domain KLF9_13_N-like superfamily was only found in *OsCNGC3* (Figure 1c). The structure of *OsCNGCs* was analyzed, and the number of UTRs and CDS was identified for each gene. The CDS number ranged from 3 to 13, and *OsCNGC12* and *OsCNGC13* had more CDSs and a unique structural domain (Figure 1a).

In terms of cis-acting elements, most *OsCNGC* family members were related to light response. They also contained elements related to stress response, such as *OsCNGC3*, *OsCNGC7*, *OsCNGC8*, *OsCNGC9*, *OsCNGC13*, *OsCNGC14*, *OsCNGC15*, and *OsCNGC16*, which had drought-induced and -related MYB-binding elements (Figure 1d). 

### 2.2. Ka/Ks Ratio Analysis of OsCNGC Homologous Genes

Covariance analysis of the *OsCNGC* genes showed three pairs of genes, including *OsCNGC2* and *OsCNGC4*, *OsCNGC12* and *OsCNGC13*, and *OsCNGC15* and OsCNGC16, (Figure 2), indicating their homology to each other, and each pair was formed due to gene duplication events. To elucidate the effect of selection pressure on the evolution of the *OsCNGC* family, Ka values, Ks values, and Ka/Ks ratio of homologous genes in the *OsCNGC* family were calculated. The results showed that the Ka/Ks ratios of the three pairs of homologous genes were all much lower than 1 (Table 1), indicating that the *OsCNGC* family has experienced strong purifying selection during evolution [21]. 

### 2.3. Functional Interaction Network Analysis of OsCNGC Genes

Functional interaction network prediction of *OsCNGC* genes showed that they had the same interacting proteins, including TPC1 as biportal calcium channel protein 1, and most of the remaining proteins are members of the protein kinase superfamily (Figure 3). 

### 2.4. Expression Profiles of OsCNGC Genes in Tissues under Normal Conditions and Abiotic Stresses

*OsCNGC* gene expression in tissues under normal conditions was analyzed. *Os-CNGC4*, *OsCNGC5*, and *OsCNGC8* showed high expression in flowers; *OsCNGC7*, *OsCNGC1*, and *OsCNGC13* were highly expressed in leaves; and *OsCNGC10* and *Os-CNGC9* showed high expression in roots and stems, respectively (Figure 4a). Detailed data are shown in (Appendix A). These results suggest that tissue specificity is an important aspect of functional differentiation for *OsCNGC* genes, and the mechanism for the differential expression patterns of most *OsCNGC* genes remains unclear. Under different abiotic stresses, *OsCNGC12*, *OsCNGC11*, and *OsCNGC15* showed high expression under drought, high-temperature stress, and low-temperature stress, respectively (Figure 4b), indicating that these genes play an important role in abiotic stress response.

### 2.5. Genetic and Allelic Diversity of OsCNCG Loci in Rice Populations

From the CDS haplotype (gcHap) data from the Rice Genome Project (3KRG), we obtained the Shannon fairness EH (Appendix A). The average gcHaps, main gcHaps, and EH of the 16 *OsCNGC* genes were 134.4, 8.4, and 0.222, respectively. However, there were large variations in genetic diversity among different *OsCNGC* genes. *OsCNGC15* had an EH of 0.491 and 484 gcHaps (Appendix A), followed by *OsCNGC5* with an EH of 0.438 and 569 gcHaps. *OsCNGC6* had the lowest EH of 0.021 and eight gcHaps (Appendix A). The CNGC gene diversity also varied considerably across rice populations. The mean EH of the 16 *OsCNGCs* was 0.187 in the Xian population, 0.155 in the Geng population, 0.206 in the Aus population, and 0.267 in the Bas population. The average and major gcHaps were 81.7 and 5.5, 39.5 and 4.4, 6.8 and 4.3, and 12.3 and 4.1 in Xian, Geng, Aus, and Bas populations, respectively (Appendix A), indicating that the frequency of gcHaps in *OsCNGC* genes is low and rare in most populations.

To understand the variations of *OsCNGC* genes among major rice populations, we estimated the genetic identity of Nei (INei) and gcHap data for the 16 polymorphic *OsCNGC* genes among all pairs of groups. *OsCNGC2*, 3, 4, 5, 8, 12, 15 and 16 showed strong XI-GJ differentiation (INei ≤ 0.2) specificity (Figure 5, Appendix A). In addition, the 16 *OsCNGC* genes showed strong differentiation in other populations. For example, *OsCNGC2* also showed strong differentiation in *Aus-XI*, *Aus-GJ*, *Aus-Bas*, and *GJ-Bas* (Figure 5, Appendix A). It can be inferred that allelic variation at the *OsCNGC* loci contributes significantly to the differentiation of major rice populations and the adaptation of different populations to the environment.

### 2.6. Impact of Modern Breeding on gcHap Diversity of OsCNGC Genes

To understand the impact of modern breeding on the gcHap diversity of *OsCNGC* genes in recent decades, we conducted a study of modern varieties (MVs) and local varieties (LANs), including 732 Xian local varieties (LANs-Xian), 358 Xian modern varieties (MVs-Xian), 328 Geng local varieties (LANs-Geng), and 139 modern varieties of Geng (MVs-Geng) (Appendix A). In the Xian population, the mean EH of the 16 *OsCNGC* genes in MVs-Xian was 0.249, which was increased by 14.1% compared with the 0.188 of LANs-Xian (Table 2). Nine *OsCNGC* genes (*OsCNGC3*, *5*, *7*, *8*, *9*, *11*, *14*, *15*, and *16*) showed increased diversity in MVs-Xian genes (Appendix A). Interestingly, MVs-Xian had an average of 31.6 gcHaps per locus, which was lower than 40.5 gcHaps per locus of LANs-Xian. A closer inspection revealed that on average, 26 gcHaps/per locus were missed in MVs-Xian relative to that in LANs-Xian (Table 2, Figure 6a), which is apparently caused by genetic bottlenecks during modern breeding (Table 2, Figure 6a). To understand the variations of *OsCNGC* genes among major rice populations, we estimated the genetic identity of Nei (INei) and gcHap data for the 16 polymorphic *Os-CNGC* genes among all pairs of groups. *OsCNGC2*, *3*, *4*, *5*, *8*, *12*, *15* and *16* showed strong *XI-GJ* differentiation (INei < 0.35) specificity (Figure 5, Appendix A). In addition, the 16 *OsCNGC* genes showed strong differentiation in other populations. For example, *OsCNGC2* also showed strong differentiation in *Aus-XI*, *Aus-GJ*, *Aus-Bas*, and *GJ-Bas* (Figure 5, Appendix A). It can be inferred that allelic variation at the OsCNGC loci contributes significantly to the differentiation of major rice populations and the adaptation of different populations to the environment.

In addition, the proportion of newly emerged gcHaps observed in modern varieties was significantly lower than that of lost gcHaps (Figure 6b,c). MVs-Xian gained an average of 17.1 new gcHaps per locus absent in LANs-Xian. The new gcHaps were evidently generated by intragenic recombination during breeding. The increase in diversity and decrease in gcHap number observed at many loci suggest that the Xian population underwent significant frequency changes in the major gcHaps during the breeding process. In fact, significant changes in the frequency of major gcHaps, F(P) were observed at nine out of the sixteen *OsCNGC* loci, including a significant decrease at eight *Os-CNGC* loci and a significant increase at one *OsCNGC* locus (Appendix A).

In the Geng population, the average EH of the 16 *OsCNGC* genes was 0.19 in MVs-Geng and 0.17 in LANs-Geng (Table 3). However, significant increases in diversity were observed only at two *OsCNGC* loci (*OsCNGC4* and *OsCNGC9*) in MVs-Geng (Appendix A). MVs-Geng had a lower mean value (13.1 gcHaps per locus) than that of LANs-Geng (23.5 gcHaps per locus) (Table 3). In fact, on average, 6.3 new gcHaps per locus were newly gained, and 16.7 gcHaps per locus were lost in MVs-Geng relative to those in LANs-Xian (Table 3). These new gcHaps were apparently generated by intragenic recombination during breeding. Significant shifts in F(P) were observed at 6 out of the 16 *OsCNGC* loci, including significant decreases in F(P) at *OsCNGC4*, *OsCNGC7*, *OsCNGC9*, and *OsCNGC15* and significant increases in F(P) at *OsCNGC6* and *OsCNGC8* (Appendix A). Notably, the same major gcHaps were found on all OsCNGC genes except for *OsCNGC2* and *OsCNGC16*, suggesting that the major gcHaps function simultaneously in Xian and Geng.

### 2.7. Association Analysis of Major gcHaps of OsCNGC Genes with Important Agronomic Traits

To further demonstrate the functional importance of the 16 CNGC genes, we constructed a gcHap network of the primary alleles of the 16 genes in 5 rice populations and analyzed their associations with four agronomic traits, including number of spikes per plant (PN), spike length (PL), plant height (PH), and thousand-grain weight (TGW) (Figure 7 and Appendix A). Strong (*p* < 10^−7^) associations were observed in 39 (60.9%) out of 64 (16 × 4) instances, and many of the major alleles of *OsCNGC* genes were strongly associated with more than one trait. We further analyzed four cloned *OsCNGC* genes, among which *OsCNGC9* is a conserved gene with six major gcHaps. Hap4 had the highest frequency in LANs-Xian (Figure 7). Hap2 differed from Hap4 by two non-synonymous mutations. Hap2 has become the major gcHap in LANs-Geng and has been significantly reduced by selection during breeding, which significantly reduced the PN. *OsCNGC13* was conserved and had two major gcHaps. Hap2 had eight non-synonymous mutations from Hap1, and selection during breeding reduced the PN, PL, PH, and TGW. *OsCNGC14* belonged to “other genes” and had six major gcHaps. Hap2 had the highest frequency in LANs-Xian, and Hap4 had the highest frequency in LANs-Geng. Hap1 showed a decline relative to Hap4 for all three traits except for PL (no change), while Hap2 showed an increase in PN. *OsCNGC16* was a divergent gene between Xian and Geng and had a staggering number of major haplotypes (11). Figure 7 clearly shows that significant differentiation occurred after four non-synonymous mutations between Hap10 and Hap4. Among them, Hap6 showed a clear dominance in PN. Hap2 and Hap9 had the highest frequency in LANs-Xian and LANs-Geng, respectively. Hap2 showed an increase only in PN compared with Hap9. In addition, the proportion of newly emerged gcHaps observed in modern varieties was significantly lower than that of lost gcHaps (Figure 6b,c). MVs-Xian gained an average of 17.1 new gcHaps per locus absent in LANs-Xian. The new gcHaps were evidently generated by intragenic recombination during breeding. The increase in diversity and decrease in gcHap number observed at many loci suggest that the Xian population underwent significant frequency changes in the major gcHaps during the breeding process. In fact, significant changes in the frequency of major gcHaps, F(P) were observed at nine out of the sixteen *OsCNGC* loci, including a significant decrease at eight *OsCNGC* loci and a significant increase at one *OsCNGC* locus (Appendix A). 

### 2.8. Comparison of Trait Values for OsCNGC Gene Dominance and Unfavorable gcHaps in Rice

The major gcHaps (with the highest frequency) of the *OsCNGC* loci in a rice population are thought to have been favored by natural selection during evolution. On the contrary, the major gcHaps with the lowest frequency in a population are likely unfavorable gcHaps [9]. We compared the phenotypic differences between major gcHaps and unfavorable gcHaps at each of the 16 CNGC loci for 15 agronomic traits. Out of the 240 (16 × 15) comparisons, significant differences between major gcHaps and unfavorable gcHaps at all *OsCNGC* loci were detected in 100 (41.7%) comparisons (Appendix A; Appendix A). TGW was detected at 10 (62.5%) *OsCNGC* loci, and it was hypothesized that these genes might have a greater effect on grain weight (Appendix A). These ten genes, except for *OsCNGC8*, also had certain expression in seeds under normal conditions (Appendix A). Among the 16 *OsCNGC* loci, significant differences in mean trait values between major gcHaps and unfavorable gcHaps were detected for *OsCNGC1* (8 traits), *OsCNGC4* (8 traits), *OsCNGC5* (8 traits), *OsCNGC6* (9 traits), and *OsCNGC9* (8 traits), and for more than half of the traits, significant differences were observed in the mean values between major gcHaps and unfavorable gcHaps. Interestingly, *OsCNGC4*, *OsCNGC5*, and *OsCNGC6* were all located on branch II of the phylogenetic tree (Figure 1b), suggesting that these five *OsCNGC* genes probably affect more traits than other *OsCNGC* loci (Appendix A).

### 2.9. Mining of Favorable Alleles in OsCNGC Loci for Yield Improvement

We found significant differences between major gcHaps in their effects on important agronomic traits at most rice loci. Therefore, we hypothesized that favorable gcHap(s) for traits to increase productivity are present at most *OsCNGC* loci, even though the favorable gcHap(s) may differ in different target environments. Therefore, there should be one or more favorable alleles defined as gcHap(s) associated with high productivity traits. The frequency of these most favorable gcHaps in MVs, and different rice populations would be of particular interest to rice breeders. Figure 8 shows the frequencies of favorable gcHaps for five traits, including PL, CN, GL, GW, and TGW, in the cloned *OsCNGC* genes in MVs-Xian and MVs-Geng as well as 3 KRG rice populations. The frequencies of favorable alleles varied greatly across different yield traits, *OsCNGC* loci, MVs-Xian and MVs-Geng, and rice populations. For example, Hap7 and Hap5 at *OsCNGC16* were associated with the highest GW and TGW, respectively, in the 3KRG germplasm, which was fixed in MVs-Xian but present in low frequency in MVs-Geng. This is consistent with the empirical observation in rice breeding that very high values for any particular yield component do not necessarily lead to the highest productivity, as yield-related traits are quantitative, not qualitative. In addition, for specific traits, new potential alleles can be identified from natural germplasm from natural habitat-oriented species or species grown in unfavorable or isolated environments, and wild infiltration can also be helpful. This result also suggests that the favorable alleles for different yield traits may be very different in the two rice subspecies or genetic backgrounds and may also differ in different environments. We also found that *OsCNGC16*-favorable gcHaps were absent in MVs-Geng for PL and PN, which were present in all four traits. These results suggested that some favorable gcHaps function only in specific subpopulations. Therefore, there is a great potential to improve yield traits and productivity of MVs by aggregating the missing favorable gcHaps.

## 3. Discussion

With reference to previous studies [3,4], sixteen *OsCNGC* genes were identified in this study, which are distributed on eight chromosomes except for chromosomes 7, 8, 10, and 11. Evolutionary analysis classified these genes into four major groups (I, II, III, and IV) and two subgroups (IV-A and IV-B). Gene structure analysis showed that the number of introns in *OsCNGCs* ranged from 3 to 13, and the number of exons varied considerably among *OsCNGC* genes. Structural domain analysis showed that all 16 *OsCNGC* genes contained the CAP_ED structural domain. *Os06g0527300*, which was considered to contain no CAP_ED structural domain [3], was retained because this domain is present in this gene as indicated by our conserved domain analysis (Figure 1c). Moreover, we identified no *LOC_Os06g33610* gene. Hence, we excluded it and renamed Os06g0527300 as *OsCNGC3.* This result is consistent with the recently published gene family visualization results [22]. We hypothesize that the reason for this phenomenon may be the different sources of the reference genome. According to the cis-acting element analysis, the *OsCNGC* genes play important roles in normal plant development and response to adverse stress, among which *OsCNGC9*, *OsCNGC13*, *OsCNGC14*, and *OsCNGC16* have been validated [5,8,12,13]. Functional interaction network analysis showed that all *OsCNGCs* contain TPC1. AtTPC1 is the first TPC channel cloned from plants, which is localized to the vesicular membrane and responsible for generating slow vesicular (SV) currents. AtTPC1 is a non-selective cation channel that can permeate through a wide range of monovalent cations and Ca2+ and potentially plays an important role in regulating the cytosol ion concentration [23]. Protein kinases are involved in stress response in plants [24]. Considering the high expression of some *OsCNGCs* in tissues under normal conditions and stress, we hypothesized that these *OsCNGCs* may play a key role in the corresponding phenotype and stress response (Figure 4).

*OsCNGC* genes may have played important roles in rice subspecificity and population differentiation during rice evolution, and there are strong subspecificity and population differentiation at most rice *OsCNGC* loci (Figure 5, Appendix A). Many *OsCNGC* genes may play an important role in rice improvement by significantly changing the diversity of major gcHaps at *OsCNGC* loci (Figure 5 and Figure 6, Table 2 and Table 3), as well as the agronomic traits associated with major gcHaps (Figure 7 and Figure 8, Appendix A). Because *OsCNGC* genes act as signaling receptors, they are expected to play important regulatory roles in many processes of rice growth and development and response to environmental cues. However, how different *OsCNGC* genes are involved in these important processes at the molecular level remains to be further elucidated.

Unexpectedly, we observed that most *OsCNGC* loci have increased diversity in MVs. However, due to manual selection, it may result in a lower MV. For example, in both selective breeding and breeding programs, selections are mostly associated with high productivity, including more compact plant types, larger or more panicles, appropriate grain weight or more tillers, and greater tolerance to biotic and abiotic stresses. In addition, genetic bottlenecks due to the relative fixity of germplasm resources used for breeding have reduced genetic diversity at most loci. Indeed, we observed an average loss of 26 (45%) gcHaps at the 16 *OsCNGC* loci during Xian breeding, while an average of 17.1 (29%) new gcHap loci appeared in MVs-Xian (Figure 6). We also found no increase or deletion of gcHaps related to *OsCNGC8* in both Xian and Geng, and it remains to be investigated whether this is an allele fixed in the variety. During the breeding in Geng, the average loss of gcHaps at the 16 CNGC loci was larger, which reached about 56%. At the same time, an average of 6.3 new gcHaps per locus (21%) appeared in MVs-Geng. Thus, the loss of gcHaps due to the genetic bottleneck effect is more significant in MVs-Geng than in MVs-Xian during modern breeding (Table 2 and Table 3). We hypothesize that this is because interline crosses are generally performed in the breeding process, which greatly increases the recombination of genes relative to the low heterosis rate of local varieties. We found that the frequency of major gcHaps in MVs-Xian and MVs-Geng was significantly lower at specific *OsCNGC* loci, suggesting that the gcHaps favored by natural selection were not artificially selected by modern breeding. The reason for this phenomenon may be that the modern breeding process is mainly targeted at yield and some alleles for adaption to natural environmental changes are abandoned; however, it will be costly to mine and use these environment-adapting genes. This also means that many rare gcHaps may be of great value in rice improvement. In summary, previous modern breeding activities have had a greater impact on *OsCNGC* loci than previously estimated [11], suggesting that *OsCNGC* genes are important in rice improvement.

In conclusion, our results suggest that natural variations at most *OsCNGC* loci have potential value for improving the productivity and tolerance of rice to abiotic stresses. All *OsCNGC* genes have potential value for rice trait improvement. Among them, *OsCNGC9*, *OsCNGC13*, *OsCNGC14* and *OsCNGC16* have been cloned [5,6,7]. Also, *OsCNGC11*, *OsCNGC12*, and *OsCNGC15* may be used to improve rice tolerance to abiotic stresses, while other genes may be used to improve rice yield traits. However, since most of the *OsCNGC* genes identified in this study had significant interactive effects on yield traits, it remains a challenge to identify favorable alleles at specific *OsCNGC* loci and compare the dominant and unfavorable allelic traits to improve specific yield traits of rice. Therefore, more efforts are needed to obtain the required information, and it is expected that some alleles that are already present or fixed in certain varieties will be identified for future application to improve rice productivity.

## 4. Materials and Methods

### 4.1. Identification and Physicochemical Characterization of OsCNGC Genes

To identify all CNGC genes in the whole genome of rice, we first downloaded the whole genome data of rice (Oryza sativa Geng/japonica), the protein sequences of Arabidopsis CNGCs in the database (https://www.ncbi.nlm.nih.gov/, (accessed on 26 September 2023)), and the Hidden Markov Models of CNGCs in the pfam (http://pfam.xfam.org/ (accessed on 26 September 2023)). We searched (Perl script) for NBD (Cyclic Nucleotide Binding Domain, pfamID:PF00027) and TM/ITP (Transmembrane or Ion Transport Domain, pfamID:PF00520) using Hm [25], which were compared in Tbtools to take the intersection, and genes containing these two structural domains were considered as members of the CNGC candidate gene family [26], and were published in the pfam (http://pfam.xfam.org/ (accessed on 26 September 2023)), NCBI (http://www.ncbi.nlm.nih.gov/cdd/ (accessed on 26 September 2023)), and SMART website (http:smartbl de/(accessed on 26 September 2023)), as well as the rice database (https://ricedata.cn/gene/ (accessed on 26 September 2023)) to validate the candidate genes of *OsCNGCs*. Moreover, by analyzing the CNGC-related literature for naming [3,4], we finally identified all the *OsCNGC* genes. Physicochemical properties such as molecular weight and isoelectric point of *OsCNGC* genes were predicted using the online tools ExPASy (https://www.expasy.org/ (accessed on 26 September 2023)) and Rice Database (https://ricedata.cn/gene/ (accessed on 26 September 2023)). Subcellular localization prediction was performed using the online tool Cell-PLoc 2.0 (http://www.csbio.sjtu.edu.cn/bioinf/euk-multi-2/ (accessed on 26 September 2023)).

### 4.2. Gene Structure, Conserved Motif, Conserved Structural Domain, and Cis-Acting Element Analysis of OsCNGC Proteins

After downloading the gene structure annotation files from the NCBI website, the gene structure view (advanced) function in TBtools-II v2.025 software was used for visualization. The conserved analysis of CNGC protein sequences was achieved using the TBtools-II v2.025 software Simple MEME Wrapper Function Implementation Function, which set 20 motifs. The number of occurrences of motifs on each sequence was unlimited, and the other parameters were the default parameters. The conserved structural domains were analyzed using the CD search function of NCBI and visualized using Tbtools [26]. Gtf/Gff3 sequence extracts from TBtools were utilized for extracting the upstream 2000 bp of CDS and extracting the promoter sequences of CNGC genes, which were submitted to the PlantCARE database (http://bioinformatics.psb.ugent.be/webtools/plantcare/html/ (accessed on 26 September 2023)) to analyze their promoter region cis-acting elements [27]. The resultant files were filtered based on the information in the table, retained for viewing, and then presented visually using the Simple BioSequence Viewer feature of TBtools-II v2.025 software.

### 4.3. Chromosomal Localization, Covariance Analysis, and Functional Interaction Network Analysis of OsCNGC Genes

Rice DNA files and annotation files were downloaded from the NCBI website. Interspecific covariates were mapped using the one-step MCScanX and advanced Circos functions of TBtools [26]. Gene positions were visualized from the GTF/GFF function to map the distribution of OsCNGC genes on chromosomes. Functional interaction network analysis was performed according to the instructions on the online website (https://www.stringdb.org/, (accessed on 26 September 2023)). STRING used a spring model to generate network images. Nodes were modeled as masses and edges as springs. The nodes were used to compute the image by minimizing the “energy” of the system. 

### 4.4. Phylogenetic Tree Construction and Gene Expression Specificity Analysis of OsCNGC Genes

Phylogenetic trees of CNGC genes of Arabidopsis, rice, maize, and Populus trichocarpa were constructed using MEGA11 software (https://www.megasoftware.net/ accessed on 30 October 2023). The full-length sequences of the CNGC proteins were matched to each other and analyzed phylogenetically. The protein sequences required for the construction of the evolutionary tree are presented in Appendix A. The phylogenetic tree was then embellished using the online mapping site iTOL (https://itol.embl.de/ (accessed on 26 September 2023)). The newly published gene family visualization website (https://bis.zju.edu.cn/cropgf/analysis/gene-info (accessed on 26 September 2023)) was used for heatmap visualization using Tbtools after downloading the expression data of *OsCNGC* genes in different tissues and under abiotic stresses [22,26].

### 4.5. gcHaps and Their Diversity in Modern and Local Varieties

Shannon’s fairness (EH) was used to assess the level of gcHap diversity at all *OsCNGC* loci in a given rice population or in the species as a whole [28]. Nei’s genetic identity (INei) was used to measure the genetic similarity between two populations or individuals based on their gcHap frequencies at different *OsCNGC* loci [29]. The formulas used to calculate EH and INei referred to the methodology described in a previous study [11], where *OsCNGC* genes were classified into conserved genes based on gcHap diversity and subspecies/population differentiation (EH < 0.3 in 3KRG and INei ≥ 0.8 between pairwise populations), and Xian-Geng differentiated genes (*X-G*) (EH < 0.3 in both Xian and Geng and INei ≤ 0.2 between Xian and Geng), and other genes [17]. To understand how modern breeding over the past decades has affected the gcHap diversity of *OsCNGC* genes, we collected detailed information on a total of 3010 3KRG rice materials. Of these, 732 were local varieties of Xian (indica) (LANs-Xian), 358 were local varieties of Geng (japonica) (LANs-Geng), 328 were modern varieties of Xian (indica) (MVs-Xian), and 139 were modern varieties of Geng (japonica) (MVs-Geng). First, we downloaded the major gcHaps (highest frequency in 3KRG) of each *OsCNGC* gene from RFGB and 3K webpage (https://www.rmbreeding.cn/Index (accessed on 26 September 2023)). The frequency difference of major gcHaps for each OsCNGC gene between modern varieties (MVs) and local varieties (LAN) was then calculated based on R-4.2.2 scripts [17]. Finally, the distribution of gcHaps in modern Xian and Geng varieties and their respective local varieties were compared. Missing and newly occurring gcHaps in modern Xian/Geng varieties were also analyzed. Finally, GraphPad Prism 8 software was used to plot the above data. 

### 4.6. Determination of Major gcHap Phenotypes of OsCNGC Genes

First, we collected phenotypic data on 3010 Asian-cultivated rice germplasm for 15 agronomic traits. This study examined 15 agronomic traits including cays to heading (DTH, day), plant height (PH, cm), flag leaf length (FLL, cm), flag leaf width (FLW, cm), panicle number (PN, count), panicle length (PL, cm), culm number (CN, count), culm length (CL, cm), grain length (GL, mm), grain width (GW, mm), grain length/width ratio (GLWR, ratio), thousand-grain weight (TGW, g), leaf rolling index (LRI,%), seedling height (SH, cm), and ligule length(LL, mm). Phenotypic data for 15 rice traits were downloaded from the RFGB website (https://www.rmbreeding.cn/Index/ (accessed on 26 September 2023)). Next, the major gcHaps (with frequency ≥1% in 3KRG) of all rice CNGC genes were obtained using R scripts [17]. Finally, associations of major gcHaps with these agronomic traits in different 3010 rice germplasm were realized using R scripts. Significance was calculated using one-way ANOVA and compared between major gcHaps using Tukey’s multiple comparisons. The layout of the images was conducted in Adobe Illustrator 2023 software, version 28.0. 

### 4.7. Construction of gcHap Network of OsCNGC Genes and Association Analysis between Major gcHaps and Yield Traits in 3KRG Materials

First, haplotypes (gcHaps) of rice CNGC genes were constructed using the R package pegas [14]. A network of gcHaps for each *OsCNGC* gene was generated using a statistical parsimony algorithm, which was performed with a method that first connects the most closely related haplotypes by the smallest number of mutations [11]. More detailed steps are presented in a previous study [18]. The layout of the images was performed in Adobe Illustrator 2023 software, version 28.0. 

## Figures and Tables

**Figure 1 plants-12-04089-f001:**
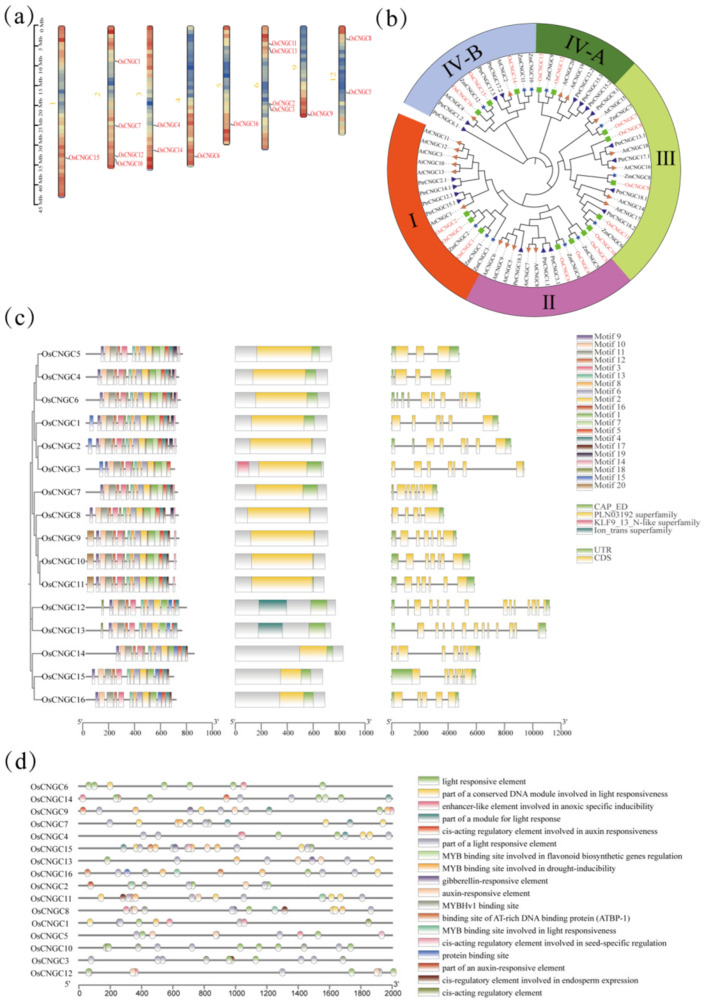
Characterization of *OsCNGC* genes. (**a**) Chromosomal localization of *OsCNGC* genes; (**b**) Phylogenetic tree of *OsCNGCs* in rice, Arabidopsis, maize, and Populus trichocarpa; (**c**) Phylogenetic tree, motif prediction, structural domains, and exon-intron structural distributions of *Os-CNGC* genes, from the left to the right, respectively; and (**d**) cis-acting elements of *OsCNGC* genes.

**Figure 2 plants-12-04089-f002:**
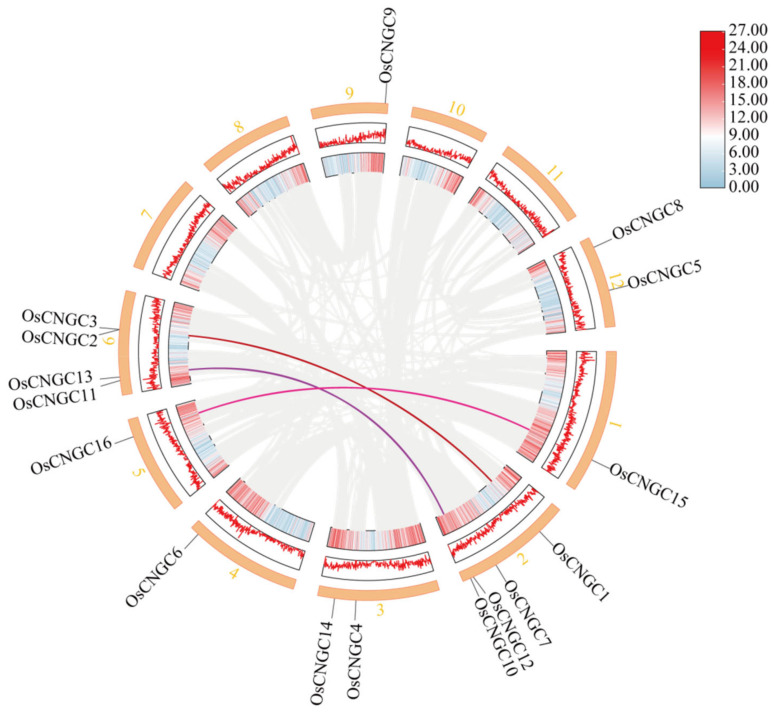
Homologous relationship and chromosomal localization of *OsCNGC* genes. The connected lines represent a homologous relationship.

**Figure 3 plants-12-04089-f003:**
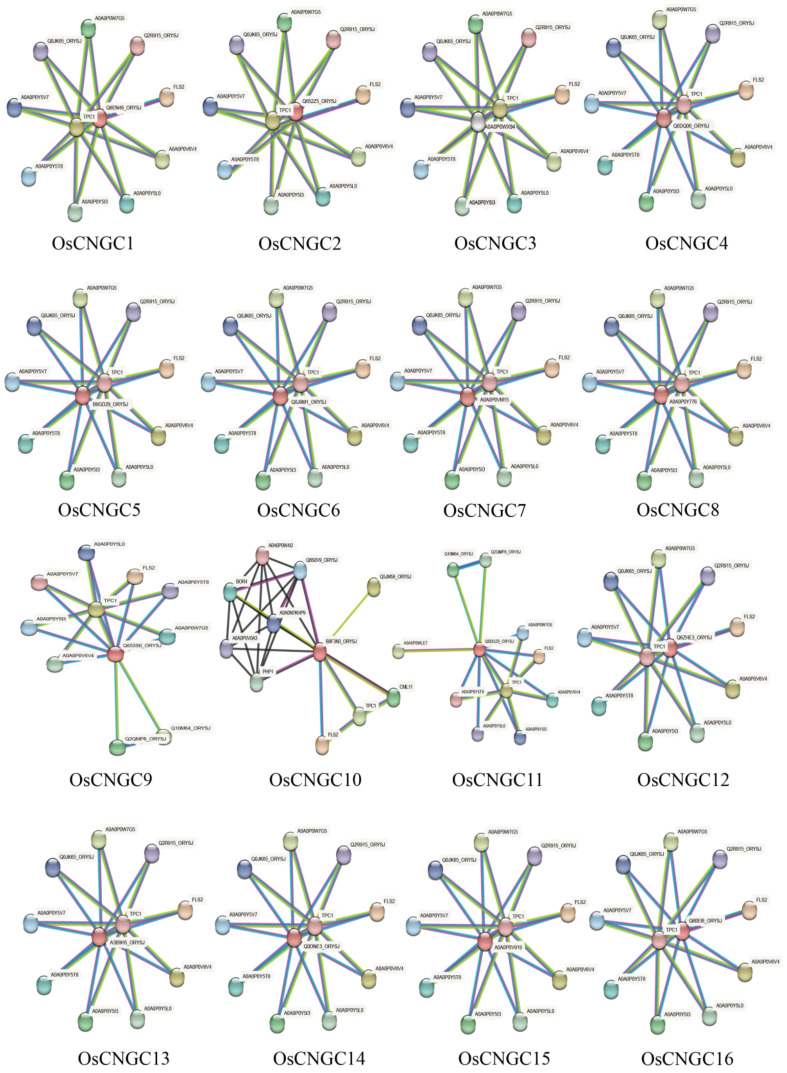
Functional interaction network of the 16 *OsCNGC* genes. The network summarizes the predicted association network for a set of specific proteins. Network nodes are proteins. Edges indicate predicted functional associations. The thickness of the line indicates the confidence prediction of the interaction. Red proteins in the center represent rice *OsCNGC* genes or their homologs. The red line indicates the presence of fusion evidence; the green line: neighborhood evidence; the blue line: co-expression evidence; the purple line: experimental evidence; the yellow line: text mining evidence; the light blue line: database evidence; the black line: co-expression evidence.

**Figure 4 plants-12-04089-f004:**
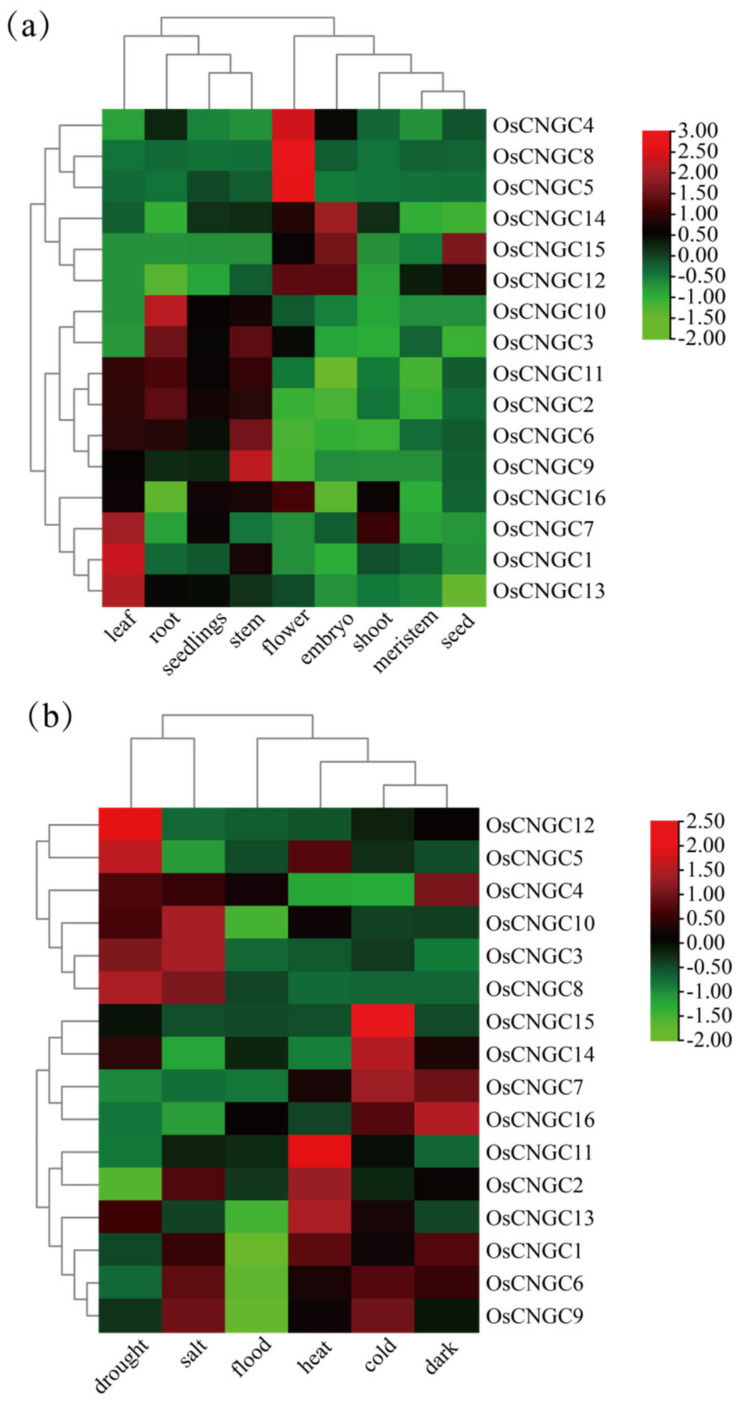
Expression analysis of *OsCNGC* genes. (**a**) *OsCNGC* gene expression in different tissues including the leaf, root, seedlings, stem, flower, embryo, shoot, meristem, and seed. (**b**) Expression of *OsCNGC* genes under abiotic stresses including drought, salt, flood, heat, cold, and dark.

**Figure 5 plants-12-04089-f005:**
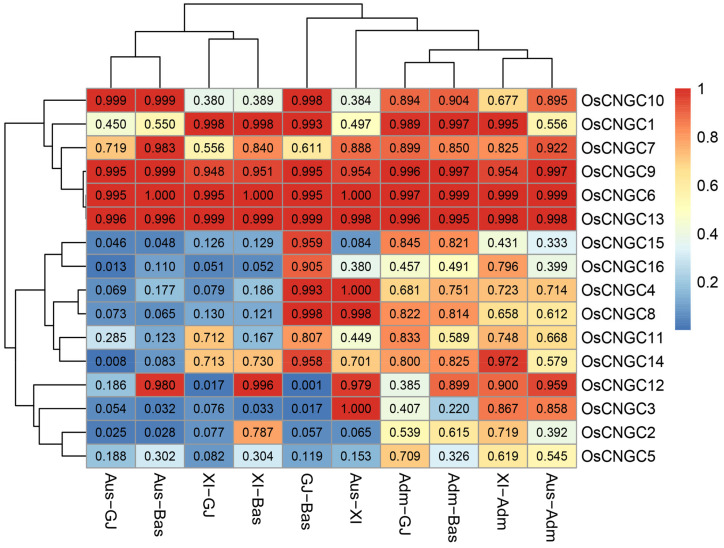
Nei’s genetic identity (INei) of *OsCNGC* genes between all paired populations calculated from gcHap data.

**Figure 6 plants-12-04089-f006:**
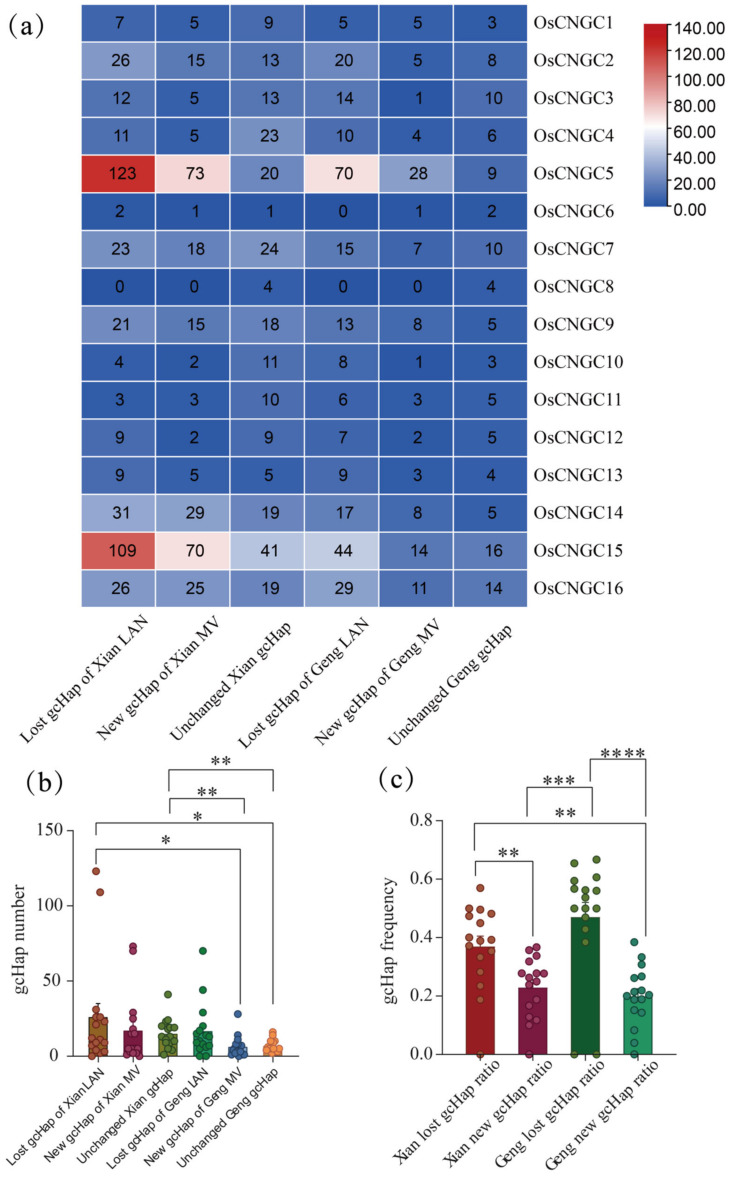
Lost, emerging, and unchanged gcHaps of the 16 *OsCNGC* genes during modern breeding. (**a**) Heat map of the number of lost, emerging, and unchanged gcHaps. (**b**) Comparative analysis of the number of lost, emerging, and unchanged gcHaps. (**c**) Frequency comparison of lost, emerging, and unchanged gcHaps. Tukey’s multiple comparison method was used. **** *p* < 0.0001, *** *p* < 0.001, ** *p* < 0.01, * *p* < 0.05. Only significant differences are shown.

**Figure 7 plants-12-04089-f007:**
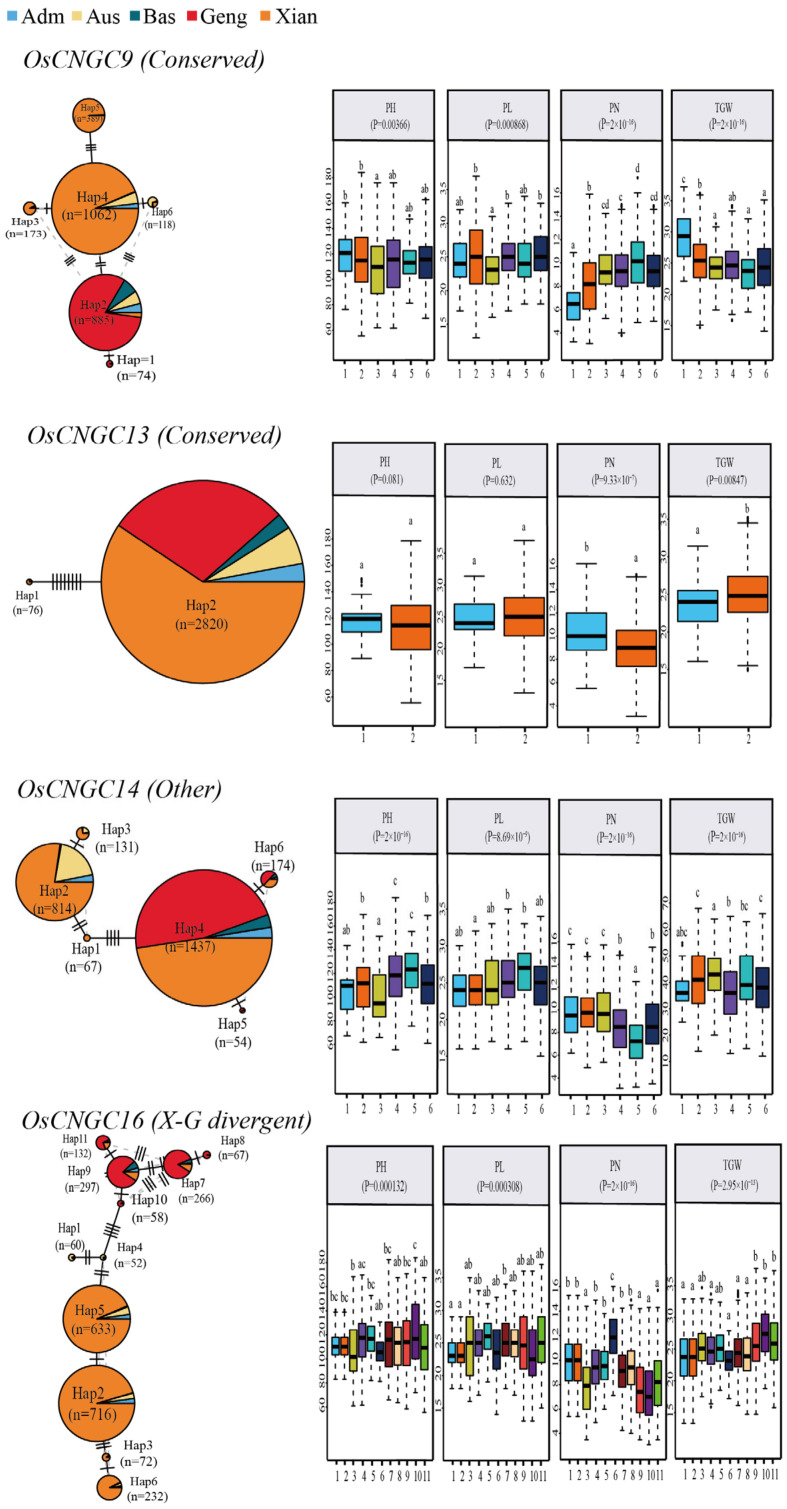
Haplotype networks of the four cloned CNGC genes and their association traits with four agronomic traits in 3KRG. *p*-Values indicate differences among haplotypes assessed by two-factor ANOVA, where different letters on the box-and-line plots indicate statistically significant differences based on the Duncan’s Multiple Range Test at *p* < 0.05. The bars on the right show the frequency differences in dominant gcHaps between local varieties (LANs) and modern varieties (MVs) in Xian and Geng. The chi-square test was used to determine significant differences in the proportions of the same gcHap between groups.

**Figure 8 plants-12-04089-f008:**
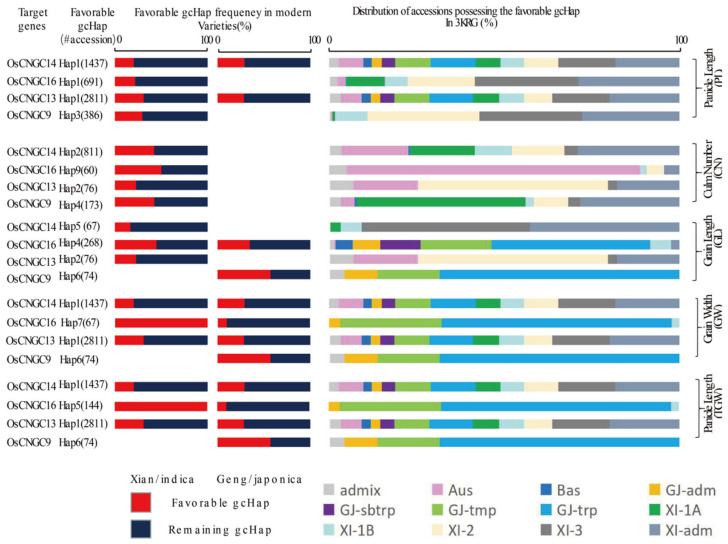
Frequency of favorable gcHaps of four cloned *OsCNGC* genes affecting five yield traits (GWT, GL, GW, PL, and CN) in modern varieties of Xian/indica (XI) and Geng/japonica (*GJ*) and different rice subgroups. “#accession” indicates the number of accessions that possess the favorable gcHap. Five subpopulations of XI (XI-1A, XI-1B, *XI-2*, *XI-3*, and *XI-adm*) and four subpopulations of *GJ* (temperate *GJ*: *GJ-tmp*, *subtropical GJ: GJ-sbtrp*, *tropical GJ: GJ-trp*, and *GJ-adm*) [18].

**Table 1 plants-12-04089-t001:** Ka/Ks ratio of *OsCNGC* homologous genes.

Gene 1	Gene 2	Ka	Ks	Ka/Ks
*OsCNGC2*	*OsCNGC4*	0.14	0.9	0.16
*OsCNGC12*	*OsCNGC13*	0.14	0.82	0.17
*OsCNGC15*	*OsCNGC16*	0.14	0.39	0.35

**Table 2 plants-12-04089-t002:** Comparison of genetic diversity of the 16 *OsCNGC* genes in Geng between local and modern varieties.

Gene Name	Xian (Indica)						
	LANs		MVs		Change of gcHapN (Major gcHapN)		
	E_H_	gcHapN (Major gcHapN)	E_H_	gcHapN (Major gcHapN)	Lost	New	Retained
*OsCNGC1*	0.13	16 (3)	0.168	14 (5)	7	5	9
*OsCNGC2*	0.196	39 (4)	0.248	28 (5)	26	15	13
*OsCNGC3*	0.153	25 (3)	0.215	18 (5)	12	5	13
*OsCNGC4*	0.111	27 (4)	0.101	21 (2)	11	5	23
*OsCNGC5*	0.38	143 (8)	0.511	93 (9)	123	73	20
*OsCNGC6*	0.003	3 (1)	0.004	2 (1)	2	1	1
*OsCNGC7*	0.307	47 (9)	0.391	42 (9)	23	18	24
*OsCNGC8*	0.07	4 (3)	0.12	4 (4)	0	0	4
*OsCNGC9*	0.21	39 (5)	0.301	33 (8)	21	15	18
*OsCNGC10*	0.143	15 (3)	0.183	13 (5)	4	2	11
*OsCNGC11*	0.157	13 (5)	0.226	13 (6)	3	3	10
*OsCNGC12*	0.056	18 (3)	0.082	11 (3)	9	2	9
*OsCNGC13*	0.056	14 (2)	0.061	10 (3)	9	5	5
*OsCNGC14*	0.26	50 (8)	0.354	48 (7)	31	29	19
*OsCNGC15*	0.502	150 (13)	0.639	111 (16)	109	70	41
*OsCNGC16*	0.271	45 (7)	0.385	44 (8)	26	25	19
Mean	0.188	40.5(5.1)	0.249	31.6 (6)	26	17.1	14.9

Note: EH and gcHapN (major gcHapN) are the fairness of the Shannon and the number of gcHaps (≥1% of varieties) identified, respectively.

**Table 3 plants-12-04089-t003:** Comparison of genetic diversity of the 16 *OsCNGC* genes in Geng between local and modern varieties.

Gene Name	Geng (Japonica)						
	LANs		MVs		Change of gcHapN (Major gcHapN)		
	E_H_	gcHapN (Major gcHapN)	E_H_	gcHapN (Major gcHapN)	Lost	New	Retained
*OsCNGC1*	0.102	8(3)	0.129	8(3)	5	5	3
*OsCNGC2*	0.171	28(6)	0.189	13(8)	20	5	8
*OsCNGC3*	0.174	24(7)	0.186	11(7)	14	1	10
*OsCNGC4*	0.07	16(3)	0.13	10(5)	10	4	6
*OsCNGC5*	0.436	79(9)	0.435	37(7)	70	28	9
*OsCNGC6*	0.045	2(2)	0.03	3(2)	0	1	2
*OsCNGC7*	0.271	25(6)	0.325	17(6)	15	7	10
*OsCNGC8*	0.084	4(4)	0.055	4(3)	0	0	4
*OsCNGC9*	0.09	18(3)	0.156	13(2)	13	8	5
*OsCNGC10*	0.068	11(2)	0.045	4(3)	8	1	3
*OsCNGC11*	0.152	11(3)	0.207	8(6)	6	3	5
*OsCNGC12*	0.046	12(1)	0.071	7(3)	7	2	5
*OsCNGC13*	0.063	13(3)	0.058	7(2)	9	3	4
*OsCNGC14*	0.177	22(4)	0.17	13(2)	17	8	5
*OsCNGC15*	0.346	60(11)	0.417	30(9)	44	14	16
*OsCNGC16*	0.421	43(11)	0.434	25(10)	29	11	14
Mean	0.170	23.5 (4.9)	0.19	13.1 (4.9)	16.7	6.3	6.8

Note: EH and gcHapN (major gcHapN) are the fairness of the Shannon and the number of gcHaps (≥1% of varieties) identified, respectively.

## Data Availability

Data is contained within the article and Appendix A.

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
