# Peer review of "Identification of the CNGC Gene Family in Rice and Mining of Alleles for Application in Rice Improvement"

_plants, 2023, doi:10.3390/plants12244089_

Round 1

Reviewer 1 Report

Comments and Suggestions for Authors

Wang et al. presented role cyclic nucleotide-gated ion channel (CNGC) gene regulation in plant immune and abiotic stress response by bioinformatic approches.  They desribed methods adequately and connected obtained results with actual knowledge about this problem. They cited 29 publication, whereare 28 were published after 2000 year. Please correct only  position number 29  in references which is incomplete. 

Author Response

Manuscript report

Wang et al. presented role cyclic nucleotide-gated ion channel (CNGC) gene regulation in plant immune and abiotic stress response by bioinformatic approches.  They desribed methods adequately and connected obtained results with actual knowledge about this problem. They cited 29 publication, whereare 28 were published after 2000 year. Please correct only  position number 29  in references which is incomplete.

  1. They cited 29 publication, whereare 28 were published after 2000 year. Please correct only  position number 29  in references which is incomplete.

Authors Response

Point-by-point responses to the reviewers’ comments:

  1. They cited 29 publication, whereare 28 were published after 2000 year. Please correct only  position number 29  in references which is incomplete.

Response: We would like to thank the reviewers for correcting the references in the article, thank you very much for your questions, which are very important to improve our writing  in the future, and thank you again for taking the time to review this article. 

Reviewer 2 Report

Comments and Suggestions for Authors

Authors presented in their research highly sophisticated molecular examinations focused on the identification of the CNGC gene family in rice . The obtained results are very important for the further rice genetic improvement. Regarding the scientific value of the research I recommend to publish it in Plants journal . Only some technical text editing errors were noticed in the text, i.e. in lines: 31, 51, 126, 205, 206, 233 "-" should be deleted in words as follows:"stress-es;  regu-late; Phyloge-netic; con-tributes; adapta-tion; adapta-tion. 

Author Response

Manuscript report

Authors presented in their research highly sophisticated molecular examinations focused on the identification of the CNGC gene family in rice . The obtained results are very important for the further rice genetic improvement. Regarding the scientific value of the research I recommend to publish it in Plants journal .

  1. Only some technical text editing errors were noticed in the text, i.e. in lines: 31, 51, 126, 205, 206, 233 "-" should be deleted in words as follows:"stress-es;  regu-late; Phyloge-netic; con-tributes; adapta-tion; adapta-tion.

Authors Response

Point-by-point responses to the reviewers’ comments:

  1. Only some technical text editing errors were noticed in the text, i.e. in lines: 31, 51, 126, 205, 206, 233 "-" should be deleted in words as follows:"stress-es;  regu-late; Phyloge-netic; con-tributes; adapta-tion; adapta-tion.

Response:We would like to thank the reviewers for their discovery of errors in the text editing errors, thank you very much for your question, which is very important to improve our writing in the future, and thank you again for taking the time to comment on this article.

Reviewer 3 Report

Comments and Suggestions for Authors

Dear Authors,

I think using online databases and based on publicly available sequences, you have done the work, and I still appreciate your rigorous input for data analysis. However, at least you could have done some wet lab work on expression analysis for a couple genotypes.

I raised some comments:

You are saying here something different from your discussion: You have mentioned "LOC_Os06g33600 as OsCNGC3" in discussion with domain analysis. It's misleading. Please address this issue.

I think a sudden increase in yield is not possible because yield-related traits are quantitative rather than qualitative. Also, for a specific trait, it's possible to identify novel potential alleles from natural germ plasm from natural habitat-oriented species or species growing in an adverse or isolated environment, and wild introgressions would also help. Hence, it's better for authors to revise their perspective.

Have obtained sequence information of cis-elements, whether there is any consistency across these genes (CNGC group) in terms of sequence structure and positions?

Seems only 6 genes or 3 pair of genes among the studied genes showed more than 80% homology. At least you should have calculated Ka/Ks for some less homologous genes as well to understand their evolutionary selection compared to homologous genes. I suppose they will be in neutral or positive selection during evolution. Hence, it is better to calculate the most homologous and least homologous genes within the same group of genes studied.

Have you performed any wet lab experiments using the genotype of interest for CNGC genes besides the Online expression database data analysis?

According to INei, you could say that in a certain population, these genes are highly conserved, while between-subspecies analysis is not, as INei<0.2 (according to method section). So, how can you infer collectively from the overall population about the homology conservation of this group of genes? Accordingly, you could rewrite wherever required.

Please correct typos, and spelling mistakes, and unnecessary punctuation marks.

If you are not sure, please don't deviate yourself from the known or presumed concepts.

You wrote as follows: "Unexpectedly, we observed that most OsCNGC loci have increased diversity in MVs, while many researchers believe that modern rice breeding has led to a decrease in MV diversity"

Accordingly, modify your following claim according to accepted rules. If not, make a hypothesis to test with modern cultivars to continue with your future perspectives on research.

Comments on the Quality of English Language

A slight English revision is required.

Author Response

Manuscript report

I think using online databases and based on publicly available sequences, you have done the work, and I still appreciate your rigorous input for data analysis. However, at least you could have done some wet lab work on expression analysis for a couple genotypes.

  1. You are saying here something different from your discussion: You have mentioned "LOC_Os06g33600 as OsCNGC3" in discussion with domain analysis. It's misleading. Please address this issue.
  2. I think a sudden increase in yield is not possible because yield-related traits are quantitative rather than qualitative. Also, for a specific trait, it's possible to identify novel potential alleles from natural germ plasm from natural habitat-oriented species or species growing in an adverse or isolated environment, and wild introgressions would also help. Hence, it's better for authors to revise their perspective.
  3. Have obtained sequence information of cis-elements, whether there is any consistency across these genes (CNGC group) in terms of sequence structure and positions?
  4. Seems only 6 genes or 3 pair of genes among the studied genes showed more than 80% homology. At least you should have calculated Ka/Ks for some less homologous genes as well to understand their evolutionary selection compared to homologous genes. I suppose they will be in neutral or positive selection during evolution. Hence, it is better to calculate the most homologous and least homologous genes within the same group of genes studied.
  5. Have you performed any wet lab experiments using the genotype of interest for CNGC genes besides the Online expression database data analysis?
  6. According to INei, you could say that in a certain population, these genes are highly conserved, while between-subspecies analysis is not, as INei<0.2 (according to method section). So, how can you infer collectively from the overall population about the homology conservation of this group of genes? Accordingly, you could rewrite wherever required.
  7. Please correct typos, and spelling mistakes, and unnecessary punctuation marks.
  8. If you are not sure, please don't deviate yourself from the known or presumed concepts.
  9. You wrote as follows: "Unexpectedly, we observed that most OsCNGC loci have increased diversity in MVs, while many researchers believe that modern rice breeding has led to a decrease in MV diversity"
  10. Accordingly, modify your following claim according to accepted rules. If not, make a hypothesis to test with modern cultivars to continue with your future perspectives on research.

Authors Response

Point-by-point responses to the reviewers’ comments: 

  1. You are saying here something different from your discussion: You have mentioned "LOC_Os06g33600 as OsCNGC3" in discussion with domain analysis. It's misleading. Please address this issue.

Response: Thank you for your comments. We have made changes in the article with the unified name Os06g0527300.

  1. I think a sudden increase in yield is not possible because yield-related traits are quantitative rather than qualitative. Also, for a specific trait, it's possible to identify novel potential alleles from natural germ plasm from natural habitat-oriented species or species growing in an adverse or isolated environment, and wild introgressions would also help. Hence, it's better for authors to revise their perspective.

Response: Thank you for your comments. We have made changes to this section of the paper based on your point of view, thank you very much for your question.

  1. Have obtained sequence information of cis-elements, whether there is any consistency across these genes (CNGC group) in terms of sequence structure and positions?

Response: Thank you for your comments. We obtained sequence information for cis elements, and these genes (CNGC group) are consistent in sequence structure and position.

  1. Seems only 6 genes or 3 pair of genes among the studied genes showed more than 80% homology. At least you should have calculated Ka/Ks for some less homologous genes as well to understand their evolutionary selection compared to homologous genes. I suppose they will be in neutral or positive selection during evolution. Hence, it is better to calculate the most homologous and least homologous genes within the same group of genes studied.

Response: Thank you for your comments. The ka/ks values we calculate are based on collinearity, and the ka/ks values cannot be calculated for genes that do not have collinearity. 80% of the homology could have been misunderstood, so we removed it.

  1. Have you performed any wet lab experiments using the genotype of interest for CNGC genes besides the Online expression database data analysis?

Response: Thank you for your comments. We have not conducted experiments in this study for the time being, but we will add experimental verification in the follow-up study.

  1. According to INei, you could say that in a certain population, these genes are highly conserved, while between-subspecies analysis is not, as INei<0.2 (according to method section). So, how can you infer collectively from the overall population about the homology conservation of this group of genes? Accordingly, you could rewrite wherever required.

Response: Thank you for your comments. Genetic distance (INei) in population analysis can be used to assess genetic diversity and kinship between different individuals or populations. In response to your question, we also found that some of the values in the article were wrong and corrected.

  1. Please correct typos, and spelling mistakes, and unnecessary punctuation marks.

Response: Thank you for your comments. We are very grateful to the reviewers for correcting the wording and punctuation that appear in the article, and we really appreciate your questions, which are very important to improve our writing skills in the future. Once again, thank you for taking the time to review this article.

  1. If you are not sure, please don't deviate yourself from the known or presumed concepts.

Response: Thank you for your comments. In response to this problem, we have redescribed the text.

  1. You wrote as follows: "Unexpectedly, we observed that most OsCNGC loci have increased diversity in MVs, while many researchers believe that modern rice breeding has led to a decrease in MV diversity"

Response: Thank you for your comments. We've rewritten this question." Surprisingly, we observed that most of the OsCNGC loci increased the diversity of mv. However, it is believed that the selection effect may lead to a decrease in the MV".

  1. Accordingly, modify your following claim according to accepted rules. If not, make a hypothesis to test with modern cultivars to continue with your future perspectives on research

Response: Thank you for your comments. We will continue to research and verify. Thank you very much for your comments.
